# Preparation of Bispecific IgY-scFvs Inhibition Adherences of Enterotoxigenic *Escherichia coli* (K88 and F18) to Porcine IPEC-J2 Cell

**DOI:** 10.3390/ijms25073638

**Published:** 2024-03-25

**Authors:** Luqing Yang, Yuanhe Yang, Anguo Liu, Siqi Lei, Pingli He

**Affiliations:** State Key Laboratory of Animal Nutrition and Feeding, Frontiers Science Center for Molecular Design Breeding (MOE), China Agricultural University, Beijing 100193, China; yang_lu_qing@163.com (L.Y.); yyh8695@163.com (Y.Y.); angelo524@foxmail.com (A.L.); leiske2000@163.com (S.L.)

**Keywords:** bispecific IgY-scFvs, inhibition, enterotoxigenic *Escherichia coli*, porcine IPEC-J2 cells

## Abstract

Enterotoxigenic *Escherichia coli* (ETEC) strains are significant contributors to postweaning diarrhea in piglets. Of the ETEC causing diarrhea, K88 and F18 accounted for 92.7%. Despite the prevalence of ETEC K88 and F18, there is currently no effective vaccine available due to the diversity of these strains. This study presents an innovative approach by isolating chicken-derived single-chain variable fragment antibodies (scFvs) specific to K88 and F18 fimbrial antigens from chickens immunized against these ETEC virulence factors. These scFvs effectively inhibited adhesion of K88 and F18 to porcine intestinal epithelial cells (IPEC-J2), with the inhibitory effect demonstrating a dose-dependent increase. Furthermore, a bispecific scFv was designed and expressed in *Pichia pastoris*. This engineered construct displayed remarkable potency; at a concentration of 25.08 μg, it significantly reduced the adhesion rate of ETEC strains to IPEC-J2 cells by 72.10% and 69.11% when challenged with either K88 or F18 alone. Even in the presence of both antigens, the adhesion rate was notably decreased by 57.92%. By targeting and impeding the initial adhesion step of ETEC pathogenesis, this antibody-based intervention holds promise as a potential alternative to antibiotics, thereby mitigating the risks associated with antibiotic resistance and residual drug contamination in livestock production. Overall, this study lays the groundwork for the development of innovative treatments against ETEC infections in piglets.

## 1. Introduction

Postweaning diarrhea (PWD) is a prevalent health issue that significantly affects piglets during the initial two-week period following weaning. This condition stems from a combination of factors, including the piglet’s immature immune system which struggles to cope with pathogens after losing passive immunity sourced from maternal milk [1]. A key culprit behind PWD is enterotoxigenic *Escherichia coli* (ETEC), a pathogenic strain that exploits this vulnerable state [2,3,4,5]. ETEC gains access to and colonizes the small intestine of piglets through specific pili or fimbriae such as K88 and F18 [6], with these adhesins playing a pivotal role in the attachment to and invasion of intestinal cells. Once established, ETEC produces potent enterotoxins heat-labile toxin (LT) and two heat-stable toxins (STa and STb) [7,8]. These toxins stimulate an excessive secretion of water and electrolytes into the intestinal lumen, leading to profuse diarrhea, dehydration, and potentially rapid death in severe cases [9,10]. During acute ETEC outbreaks, mortality rates can soar to alarming levels, reaching up to 20–30% within a matter of days among infected piglets [11]. The research conducted further underscores the significance of K88 and F18 fimbrial types, as they were detected in an overwhelming majority (92.7%) of ETEC strains associated with PWD [2]. While antibiotics are currently the mainstay for treating bacterial diarrhea therapeutically, their extensive use has raised concerns over antibiotic resistance and the potential presence of drug residues [12,13]. This pressing issue necessitates the urgent development and implementation of alternative therapeutic strategies to manage ETEC infections effectively and sustainably, thereby reducing economic losses and improving animal welfare in the swine industry.

Chicken IgY has demonstrated several advantageous properties compared to mammalian antibodies [14]. In the context of animal health, particularly with respect to enteric diseases, IgY has shown significant efficacy against various diarrheagenic pathogens including ETEC [15], *Salmonella* spp. [16], and porcine epidemic diarrhea virus [17]. However, the further application of IgY is usually limited by individual differences in chicken [18]. Recently, single-chain variable fragments (scFvs) derived from chickens have emerged as a clinically viable option due to their unique structure and functionality. ScFv is a recombinant molecule that fuses the variable domains of the heavy chain (VH) and light chain (VL) of an antibody via a flexible linker peptide [19]. ScFvs possess several attributes that make them attractive for new antibody development: they are small in size, relatively easy to purify, exhibit strong tissue penetration, can be easily constructed and expressed, and have low immunogenicity and production costs [20]. Moreover, bispecific antibodies take advantage of this single-molecule format by combining two different antigen-binding sites within one construct, offering benefits akin to cocktail therapy while overcoming some limitations of monospecific scFvs. In the case of ETEC-induced diarrhea in piglets, the use of bispecific IgY-scFvs constructs presents a potential strategy to combat the infection effectively. By harnessing both the biological advantages of IgY-scFvs and the versatility of bispecific antibodies, bispecific IgY-scFvs could potentially neutralize multiple ETEC toxins or target distinct bacterial components simultaneously, thus providing enhanced protection against postweaning diarrhea caused by ETEC. This innovative approach underscores the significance of continued research into the application of IgY-scFvs and bispecific antibodies in veterinary medicine.

The research objective was to generate and evaluate bispecific IgY-scFvs targeting ETEC. In order to evaluate bispecific scFv inhibition adherence of ETEC, scFvs specific to K88 and F18 were isolated from chicken immunized library by phage display technology. Then, bispecific scFv was constructed and expressed in *Pichia pastoris*, which could inhibit adherence of ETEC K88 and F18 to porcine IPEC-J2 cells. This work provides the foundation for the creation of novel treatments for ETEC infection.

## 2. Results

### 2.1. Fimbrial Typing of ETEC Isolates

The ETEC strains in this study were subtyped by PCR and sequencing using the primers specified in Appendix A. PCR generated the following amplicons: 484 bp for the K88 fimbria gene and 490 bp for the F18 fimbria gene (Appendix A). These analyses showed that the K88 and F18 strains were successfully isolated and purified from fecal samples of piglets suffering from diarrhea.

### 2.2. Construction of the ScFv Library against ETEC

It was verified that there was no bacterial colony developing on the LB plate, ensuring the security of the vaccinated laying hens. After six immunizations with ETEC, the titer of anti-K88 IgY reached 1:32,000 (Appendix A), and the titer of anti-F18 IgY reached 1:32,000 (Appendix A), demonstrating that the chicken had received an effective immunological response from its vaccinations for subsequent experiments.

Chicken spleen was used to extract total RNA, which was then reverse transcribed into cDNA. The VH and VL fragments, with estimated sizes of around 420 bp and 370 bp, respectively, were amplified by PCR using the primers in Appendix A (Figure 1A,B). The scFv fragment, approximately 750 bp in length (Figure 1C), was assembled by SOE-PCR using the common nucleotide sequence of a flexible linker. The enzyme-linked products were electrotransformed, resulting in an initial size of 2.12 × 10^7^ CFU/mL for the anti-K88 scFv library and 1.66 × 10^7^ CFU/mL for the anti-F18 scFv library. A total of 24 phage clones were randomly selected from each library for the determination of the scFv insertion ratio. Colony PCR analysis indicated a 100% ratio (Appendix A). The phage display libraries against the K88 and F18 screening scFvs were effectively generated as shown by the data.

### 2.3. Bio-Screening and Sequencing of ScFv against ETEC

The biopanning, elution and amplification procedures targeting K88 and F18 molecules resulted in significant enrichment of specific K88- and F18-binding phage clones, which was confirmed by the output phage titers and polyclonal phage ELISA results from the first to the fourth round (Appendix A, Appendix A). Ninety-six individual clones were chosen at random from the fourth round to assess the binding capability of K88 and F18 using indirect phage-ELISA. The results demonstrated the ability of 96 clones’ periplasmic extracts to selectively bind K88 or F18 (Figure 2). The selection process involved choosing the top five clones with the highest ratios of K88 antigen wells to CBS wells and the top five clones with the highest ratios of F18 antigen wells to CBS wells. Among the five anti-K88 scFv clones, two distinct anti-K88 scFv sequences were identified through sequencing by the GBACK primer and categorizing based on the CDRs’ amino acid sequences. Similarly, among the five anti-F18 scFv clones, two distinct anti-F18 scFv sequences were identified using the same method (Figure 3).

### 2.4. Evaluation of scFv against K88 and F18

Four plasmids encoding scFvs were successfully transfected into BL21DE3 cells. Following 16 h of induction with 0.2 mM IPTG at 28 °C, four soluble proteins with predicted sizes of roughly 26 kDa were isolated and identified by SDS-PAGE (Figure 4A,B). Furthermore, the results of the Western blot analysis demonstrated that the four distinct scFvs could react with the His-tag monoclonal antibody (Appendix A). IPEC-J2 cells serve as appropriate in vitro models for examining the adhesion of porcine ETEC [6,21]. The scFv expressed in *E. coli* was co-incubated with K88 or F18 and the results indicated that the scFv can inhibit the adhesion of ETEC to IPEC-J2 (Figure 4C,D). The specific scFv co-incubated with K88 or F18 significantly reduced adhesion to IPEC-J2 cells compared to PBS (*p* < 0.05). The inhibition rate of adhesion to K88 or F18 increased with increasing scFv concentration (*p* < 0.05). The addition of 2.51 μg, 12.54 μg, and 25.08 μg of anti-K88 scFv-1 (K88-1) inhibited K88 by 40.06%, 66.17%, and 75.26%, respectively, while anti-K88 scFv-2 (K88-2) inhibited K88 by 59.05%, 69.92%, and 79.39%, respectively. In addition, the addition of 2.54 μg, 12.70 μg, and 25.39 μg of anti-F18 scfv-1 (F18-1) inhibited F18 by 19.44%, 29.51%, and 37.70%, respectively. Similarly, the anti-F18 scFv-2 (F18-2) agent inhibited F18 expression by 20.82%, 50.67%, and 65.70%, respectively (Figure 4C,D).

### 2.5. Evaluation of Bispecific scFv Inhibition Adherence ETEC K88 and F18 to Porcine IPEC-J2 Cells

*Pichia pastoris* is a well-established heterologous expression system. The achievement of high levels of expression in *Pichia pastoris* is dependent on several factors, including codon optimization, copy number and signal peptide information [22]. Bispecific IgY-scFvs was successfully expressed in *Pichia pastoris* X33 after codon optimization. SDS–PAGE analysis of the culture supernatant of *P. pastoris* revealed a band corresponding to recombinant bispecific IgY-scFvs with a molecular mass of approximately 61 kDa (Figure 5A). The observed molecular mass is higher than the theoretical molecular mass of the fusion protein due to glycosylation in *Pichia pastoris* [23]. Western blotting also revealed that recombinant protein with a molecular mass of approximately 61 kDa was present (Appendix A), indicating that the fusion protein can specifically bind to the mouse anti-His monoclonal antibody.

Coincubation of bispecific IgY-scFvs expressed in X33 with K88, F18 or both strains showed that the bispecific single-chain antibody inhibited the adhesion of both K88 and F18 to IPEC-J2 cells (Figure 5B). The addition of 25.08 μg of bispecific single-chain antibody reduced the adhesion rate of the strain to IPEC-J2 cells by 72.10% and 69.11% in the presence of only K88 or F18, respectively, compared to PBS. When both K88 and F18 were present, the adhesion rate of the strain to IPEC-J2 cells was reduced by 57.92%.

## 3. Discussion

Phage antibody library technology enables polyclonal antibodies to be displayed on the surface of phages through phage display technology, combining genotypic and phenotypic information, combining selectivity with amplification capability, expanding screening capacity, and obtaining specific antibodies after elution and antibody activity detection [24]. This study constructed an avian-derived phage-scFv library against ETEC for the first time. The strains used in this study were isolated from piglets exhibiting typical clinical signs of PED, representing both endemic and pandemic field strains in China, to ensure antibody library quality. The final antibody library capacities were 2.12 × 10^7^ CFU/mL and 1.66 × 10^7^ CFU/mL, which are moderate results for a phage display library [25].

IgY-scFvs are a promising class of therapeutic molecules with unique advantages in targeted therapy [26,27,28]. Many fully developed IgYs were induced with ETEC due to targeted pathogen stimulation of the host immune system. Screening of IgY-scFv was possible due to its sufficient antibody library capacity and good diversity against ETEC. After four rounds of screening, we successfully screened two anti-K88 and two anti-F18 IgY-scFv strains. The CDR regions of these IgY-scFv strains exhibited significant differences. These differences in CDR diversity may indicate differences in binding modes for specific epitopes that are critical for determining specificity [29]. This is supported by the differences in the inhibition rate of ETEC adhesion to IPEC-J2 cells caused by IgY-scFv. Differences in antibody sequences are evident in VH-CDR3, the region that binds tightly to the antigen [30]. Studies have demonstrated that mutations in the arginine residues of ETEC adhesins can entirely eliminate the binding activity of ETEC to host cells [31]. Therefore, modifying VH-CDR3 to enhance its interaction with the arginine residues on ETEC may be a feasible option.

In an experiment in which specific IgY-scFv was expressed in BL21DE3 cells to inhibit the adhesion of ETEC to IPEC-J2 cells, it was found that the effect of K88-2 was superior to that of K88-1, and the effect of F18-2 was superior to that of F18-1. Thus, to construct bispecific IgY-scFvs, K88-2 and F18-2 were chosen and linked by a flexible linker ((G_4_S)_3_). Diarrhea caused by ETEC in piglets is usually not caused by one bacterial agent; therefore, therapeutic drugs that target multiple pathogens simultaneously are urgently needed. The bispecific IgY-scFvs binds to both the K88 and F18 epitopes and was found to inhibit the adhesion of K88 and F18 to IPEC-J2 cells, which was comparable to the inhibitory effects observed with the anti-K88 IgY-scFv and anti-F18-scFv. Studies have shown that IgY can protect the body from ETEC attacks by inhibiting the adhesion of ETEC to the intestinal tract [32]. This indicates that the antibody may be a potential therapeutic agent for diarrhea caused by ETEC.

In the biomedical field, the *Pichia pastoris yeast expression system* is an essential tool for antibody production that involves numerous posttranslational modifications, including proper protein folding, glycosylation and disulfide bond formation, as well as increased recombinant protein yields and purity [33,34]. The expression of the bispecific IgY-scFvs gene was regulated by the strong and tightly controlled promoter of the alcohol oxidase (A0X1) gene, which allowed the induction of gene expression by 1% methanol. In addition, the coding sequence was fused to an alpha factor pro-peptide to ensure that the recombinant protein powder was secreted into the extracellular medium. In this study, bispecific IgY-scFvs was successfully expressed in *Pichia pastoris* and purified using Ni affinity chromatography. The synthesis is efficient and the purification process is simple. This provides a solid basis for future applications.

## 4. Materials and Methods

### 4.1. Preparation of ETEC K88 and F18 Antigens

The process for preparing the antigens derived from ETEC K88 and F18 strains involved a meticulous series of steps to ensure the accurate isolation, inactivation, and safe use of these bacteria as immunogens. Firstly, the ETEC K88 and F18 strains were isolated from the fecal samples of piglets with diarrhea. This was accomplished through a combination of conventional microbiological techniques including morphological identification on selective media such as MacConkey agar and erythromycin blue agar, followed by molecular confirmation using PCR assays targeting specific fimbrial genes (K88 and F18) and 16S rRNA sequencing for species verification. The identified strains were then maintained in our laboratory for further study. To prepare the antigens, the cultures were grown under standard conditions in Luria-Bertani (LB) broth at 37 °C with agitation at 220 rpm for an overnight incubation period. Following growth, bacterial cells were harvested via centrifugation at 12,000× *g* for 10 min at 4 °C to pellet the cells and separate them from the culture medium. For complete inactivation, the bacterial pellets were treated with 0.3% formaldehyde solution to render them non-viable. This fixation step was carried out for a duration of 72 h at 37 °C to guarantee the absence of any residual metabolic activity or pathogenicity. Post-fixation, the formaldehyde was removed by washing the fixed cells three times with phosphate-buffered saline (PBS), ensuring no carry-over of the chemical agent. After thorough washing, the antigen preparations were resuspended in PBS to the same volume as the original cell suspension. To confirm successful inactivation, a plating count was performed. With no colonies detected on the plates, it was concluded that all ETEC strains had been effectively rendered dormant and could be safely used as antigens.

### 4.2. Chicken Immunization

Nine 90-day-old laying hens were randomly divided into three groups (*n* = 3) and intramuscularly vaccinated with K88, F18 or phosphate-buffered saline (PBS, pH 7.4) as a control at five different sites (0.2 mL each) in the breast muscle. For each strain, a dose of approximately 1.0 × 10^9^ CFU/mL was administered in a volume of 1 mL. Five additional vaccinations were administered after the first vaccination, with a two-week interval between each vaccination. Spleens from chickens were taken on the seventh day following the last immunization.

### 4.3. Extraction of IgY

IgY was extracted from the yolks using polyethylene glycol (PEG) following the method described with some modifications [35]. After aseptic isolation of the yolk, the yolk solution was mixed with PBS (pH 7.4) at a ratio of 1:2 (*v*:*v*), and the fat component was removed by the addition of a gradient of increasing amounts of PEG 6000 (3.5%, 8.5%, and 12% *w*/*v*) and centrifugation (12,000× *g* for 20 min at 4 °C). The resulting precipitate was resuspended in PBS and dialyzed in PBS solution overnight. In all subsequent tests, the yolk antibody solution was collected and measured using a BCA protein quantification kit.

### 4.4. Determination of the Specific IgY Titer

ELISAs were performed to monitor the titer development of IgY against K88 and F18 as in a previous study, with minor modifications [35]. Briefly, 96-well plates were coated with K88 and F18 (10^7^ CFU/well) in carbonate–bicarbonate buffer (0.05 M, pH 9.6) at 4 °C overnight. Post-coating, the plates were washed thrice with PBS containing 0.5% Tween-20 (PBST) to remove unbound antigen. Next, the non-specific binding sites were blocked by adding 250 µL of PBS containing 2.5% skim milk powder to each well and incubating the plate for 2 h at 37 °C. Following another series of three PBST washes, the plates were treated with serially diluted IgY in PBS supplemented with 2.5% skim milk powder (100 µL per well) and incubated for 2 h at 37 °C to allow for antibody–antigen binding. Subsequently, after washing the plate three times with PBST, horseradish peroxidase (HRP)-conjugated goat anti-chicken IgY (Abcam, Cambrige, UK) was applied at a dilution ratio of 1:5000 in PBS with 5% skim milk powder (100 µL per well) and incubated for 1 h at 37 °C. Once more, the plate underwent three rounds of PBST washes to eliminate any unbound secondary antibodies. The colorimetric reaction was then initiated by adding 100 µL of TMB reagent (Solaibao, Beijing, China) to each well. This mixture was allowed to incubate for 15 min at 37 °C. To halt the reaction, 50 μL of stop solution (Solaibao, Beijing, China) was added to every well. An American Biotek Synergy 4 microplate reader was used to measure the optical density (OD) at 450 nm. The IgY titer was defined as the highest dilution factor of the sample with an OD sample/OD negative > 2.1.

### 4.5. Construction of the Library and Selection of Anti-ETEC K88 and F18 scFvs

The scFvs libraries were constructed following a previous study [36]. In summary, VH and VL genes were amplified via PCR from cDNA that was inverted from spleen RNA of immunized chickens. Subsequently, the scFv gene was constructed through SOE-PCR. The restriction enzyme *Sfi I* was then used to ligate the scFv gene into the pComb3xss vector. Electroporation was used to transform the ligated products into *E. coli* ER2738. To assess the qualities of the original library, an aliquot of the electroporated cells was cultivated. A solid-phase protocol was used to select high-affinity scFv to K88 and F18. Four wells on a microtiter plate were coated with a concentration of 10^8^ CFU/mL K88 or F18 (100 μL per well), respectively. Then, two wells were coated with 100 μL of 2% bovine serum albumin (BSA) in 0.05 M CBS (pH 9.6) and then incubated at 4 °C overnight. The next day, all wells were blocked with 2% BSA in PBS (pH 7.4) at 37 °C for an hour, followed by five rounds of washing with PBS containing 0.05% Tween (PBST). One hour of incubation at 37 °C was required for the mixing of 100 µL of the phage-scFv with an equivalent amount of 3% skimmed milk in PBS. Then, two wells coated with BSA were filled with a 100 µL aliquot of the phage solution, and they were incubated for an hour at 37 °C. The next four wells coated with ETEC received an equal distribution of the solution. Following ten PBST washes, the unbound phages were eliminated. After eluting the bound phages with 100 μL of elution buffer (0.2 M glycine-HCl (pH 2.5)), the mixture was allowed to sit at room temperature for ten minutes. The next step was to counterbalance by adding 50 μL of 2 M Tris-HCl (pH 7.4). Following titration and amplification in the following selection cycle, the eluted phages were subsequently transferred to infect ER2738 at an OD_600_ of 0.5. With the bacterial coating concentrations dropping, four rounds of panning were carried out.

### 4.6. Phage ELISA

To detect positive clones, polyclonal phage ELISA was carried out after the panning, elution, and selection process, and then monoclonal phage ELISA. Following amplification, 96 distinct colonies from the 2YT plate containing the fourth round of eluted phages were chosen at random for monoclonal phage ELISA. To carry out the phage ELISA, A 96-well plate was coated with 1 × 10^8^ CFU/mL of diluted ETEC in CBS and left overnight at 4 °C. The plate was incubated for an hour at 37 °C after the monoclonal and polyclonal phages from each round were added to each well. Using an HRP-conjugated anti-M13 mouse monoclonal antibody from Sino Biological, the phage on the surface was detected. TMB reagent (Solaibao) was added to develop the color, then 50 μL of stop solution (Solaibao) was added to halt the reaction. The absorbance readings were determined with a microplate reader at 450 nm. Colonies with positive outcomes were sequenced. AbRSAs were used to delimit and number the CDR regions of all the scFv [37].

### 4.7. Expression and Purification of the Specific scFv in E. coli

For expressing scFv, competent *E. coli* BL21DE3 cells were transformed using recombinant plasmids. After that, the cells were incubated at 37 °C in LB (50 μg/mL ampicillin) medium until the OD_600_ reached roughly 0.6. Subsequently, the scFvs were induced with 0.2 mM IPTG overnight at 28 °C. Subsequently, the cell suspensions were broken by sonication and separated by centrifugation (12,000× *g*, 10 min). The supernatants were subsequently collected and filtered through a 0.22 μm membrane. A Ni-NTA Superflow Agarose column was used to purify the soluble scFvs with a 6× His tag, and PBS was used for dialyzing. SDS-PAGE (12% sodium dodecyl sulfate–polyacrylamide gel electrophoresis) was performed and the proteins were identified by WB using an anti-His antibody after being stained with Coomassie brilliant blue to determine their purity. BCA protein assay kit was also used to assess the scFv concentrations.

### 4.8. Specific scFv Adherence Inhibition Assay

The protocols previously reported were followed to perform the scFv adherence inhibition experiment [6]. ETEC bacteria (5 × 10^5^ CFU) were treated with scFv or PBS and then incubated together for 30 min at room temperature on a shaker set at 50 rpm. A confluent monolayer of IPEC-J2 cells was cultured in each well of a 24-well tissue culture plate, and each mixture of scFv/bacteria (final volume: 600 μL) was added. After an hour of incubation at 37 °C in an incubator with 5% CO_2_, the 24-well plates were carefully cleaned with sterile PBS to get rid of any nonadherent bacteria; 0.5% Triton X-100 was used for the lysis process. The lysates that contained ETEC bacteria were then serially diluted and plated onto LB plates. The quantity of adherent bacteria (CFU) on the intestinal epithelial cells was determined after overnight incubation at 37 °C.

### 4.9. Construction of Bispecific scFv

To maintain the independence of the protein folding and function, the flexible (Gly_4_Ser)_3_ amino acid sequence, which has been widely used in domain ligation, was adopted to link the C terminal of K88-2 and the N terminal of F18-2. The gene sequence was optimized based on the codon preference of *Pichia pastoris*. The optimized gene sequence was synthesized by Nanjing Kingsley and ligated to *EcoR I* and *Xba I* in pPICZαA. Electroporation was used to transform the *Pichia pastoris* strain X33. The plasmid (2 μg) was linearized with *Sac I*, recovered using an omega cycle purity kit, dissolved in 10 μL of ddH_2_O, and transformed by electroporation of X33 at 2000 V and 5 ms. The transformants were incubated in yeast extract peptone dextrose Sorbitol (YPDS, 100 μg/mL Zeocin) plate medium at 28 °C for three days and subsequently incubated in yeast extract Peptone Dextrose (YPD, 100 μg/mL Zeocin) plate medium at 28 °C for 24 h. Next, a single clone was selected and grown in YPD (100 μg/mL Zeocin) liquid medium at 28 °C and 250 rpm for 24 h. The plants were subsequently transferred to 50 mL of buffered minimal glycerol medium and grown to the logarithmic phase. The cells were collected and resuspended in buffered minimal methanol medium. The expression of bispecific scFv was induced by adding 100% methanol to a final concentration of 1% (*v*/*v*) every 24 h. After 72 h, the cells were concentrated at 5000 rpm for 5 min, after which the supernatant was collected for protein purification and cell adhesion inhibition assays, as described in Section 4.8.

### 4.10. Statistical Analysis

All the assays were independently repeated at least three times. One-way analysis of variance (ANOVA) and Student’s *t*-test were used to compare the results. The data were analyzed using GraphPad Prism version 7.0 (GraphPad Software, San Diego, CA, USA).

## 5. Conclusions

In conclusion, bispecific IgY-scFvs were generated in *Pichia pastoris* strain X33 to target ETEC K88 and F18. The scFv was found to inhibit ETEC adhesion to small intestinal epithelial cells in vitro. By targeting and impeding the initial adhesion step of ETEC pathogenesis, this antibody-based intervention holds promise as a potential alternative to antibiotics, thereby mitigating the risks associated with antibiotic resistance and residual drug contamination in livestock production. Overall, these findings could lead to the development of scFv-based drugs to treat and prevent ETEC-induced diarrhea.

## Figures and Tables

**Figure 1 ijms-25-03638-f001:**
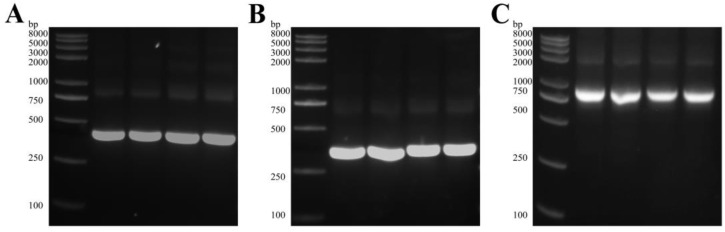
Agarose gel electrophoresis of PCR-amplified products of variable gene fragments. (**A**) PCR amplification of VH; lane 1: DNA molecular weight marker, lane 2 and 3: VH of anti-K88 IgY, lane 4 and 5: VH of anti-F18 IgY. (**B**) PCR amplification of VL; lane 1: DNA molecular weight marker, lane 2 and 3: VL of anti-K88 IgY, lane 4 and 5: VL of anti-F18 IgY. (**C**) Construction of scFvs by SOE PCR; lane 1: DNA molecular weight marker, lane 2 and 3: scFv of anti-K88 IgY, lane 4 and 5: scFv of F18 IgY.

**Figure 2 ijms-25-03638-f002:**
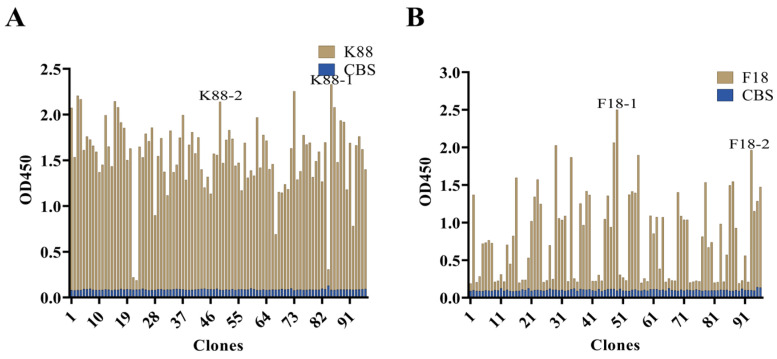
Characterization of phage displayed anti-K88 and -F18 scFvs. (**A**) Screening scFvs against k88. (**B**) Screening scFvs against F18.

**Figure 3 ijms-25-03638-f003:**
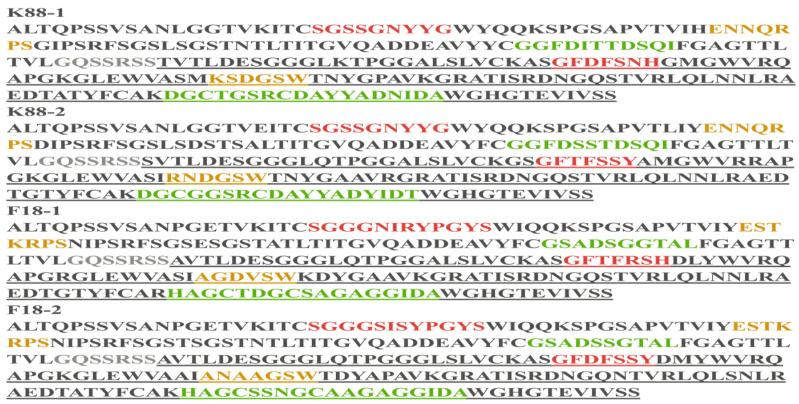
Delimitation CDRs in positive scFvs were performed with AbRSA. CDRs are highlighted in colors (**CDR1**, **CDR2**, **CDR3**). The gray letters indicate the non-variable-domain region. The underlined black letters indicate variable domain of heavy chain while the other black letters indicate variable domain of light chain.

**Figure 4 ijms-25-03638-f004:**
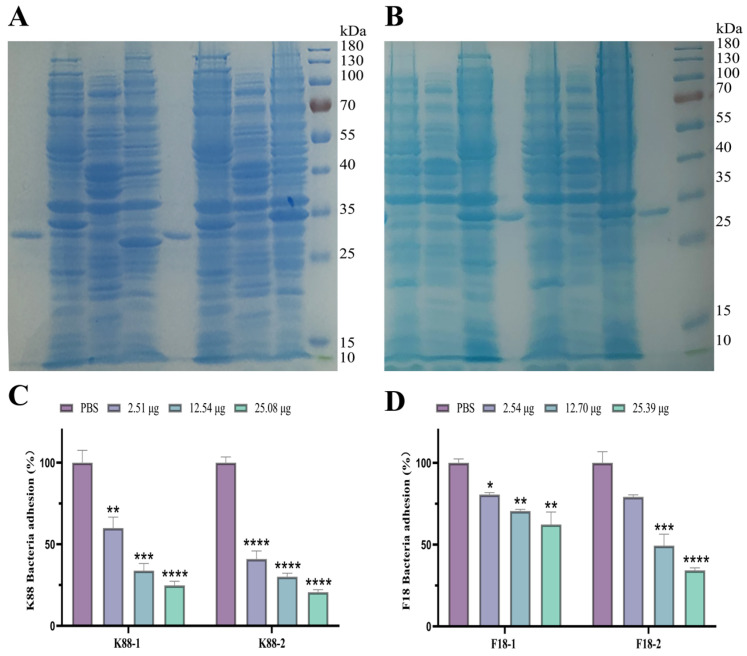
Expression and characterization of the specific IgY-scFv in *E. coli*. (**A**) Expression and purification of the specific anti-K88 IgY-scFv in *E. coli*; lane 1: purified anti-k88 IgY-scFv 1 (K88-1), lane 2: precipitation of an extract of expressing K88-1 bacteria after ultrasonic fragmentation, lane 3: supernatant of the expressing K88-1 bacteria after ultrasonic fragmentation, lane 4: no IPTG induction, lane 5: purified anti-k88 IgY-scFv 2 (K88-2), lane 6: precipitation of an extract of expressing K88-2 bacteria after ultrasonic fragmentation, lane 7: supernatant of the expressing K88-2 bacteria after ultrasonic fragmentation, lane 8: no IPTG induction, lane 9: protein molecular weight marker. (**B**) Expression and purification of the specific anti-F18 IgY-scFv in *E. coli*; lane 1: no IPTG induction, lane 2: supernatant of the expressing F18-1 bacteria after ultrasonic fragmentation, lane 3: precipitation of an extract of expressing F18-1bacteria after ultrasonic fragmentation, lane 4: purified anti-F18 IgY-scFv 1 (F18-1), lane 5: no IPTG induction, lane 6: supernatant of the expressing F18-2 bacteria after ultrasonic fragmentation, lane 7: precipitation of an extract of expressing F18-2 bacteria after ultrasonic fragmentation, lane 8: purified anti-F18 IgY-scFv 2 (F18-2), lane 9: protein molecular weight marker. (**C**) Anti-k88 IgY-scFv inhibited adherence of porcine K88 strains to porcine small intestinal cell lines (IPEC-J2). (**D**) Anti-F18 IgY-scFv inhibited adherence of porcine F18 strains to porcine small intestinal cell lines (IPEC-J2). The number of adherent bacteria (CFU) was converted to percentages, with CFU from cells treated with PBS set to 100%. The results are expressed as mean standard deviation of at least three independent experiments in three wells. Significant differences are reported with * indicating a *p*-value: 0.01 < *p* < 0.05, ** indicating a *p*-value: 0.001 < *p* < 0.01, *** indicating a *p*-value: 0.0001 < *p* < 0.001, and **** indicating a *p*-value: *p* < 0.0001.

**Figure 5 ijms-25-03638-f005:**
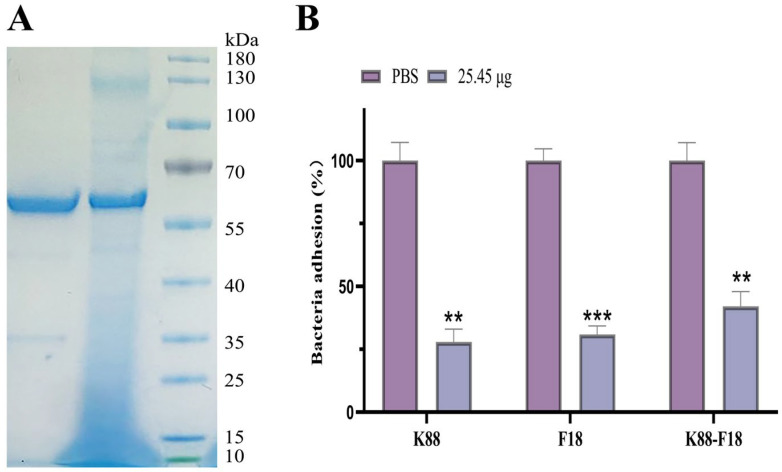
Expression and characterization of the bispecific IgY-scFvs in the *Pichia pastoris* strain X33. (**A**) Expression and purification of the bispecific IgY-scFvs in the *Pichia pastoris* strain X33; lane 1: purified bispecific IgY-scFvs, lane 2: total protein fraction from the methanol-induced strain, lane 3: protein molecular weight marker. (**B**) Bispecific IgY-scFvs inhibited adherence of ETEC K88 and F18 strains to IPEC-J2. The number of adherent bacteria was converted to percentages, with CFU from cells treated with PBS set to 100%. The results are expressed as mean standard deviation of at least three independent experiments in three wells. Significant differences are reported with ** indicating a *p*-value: 0.001 < *p* < 0.01, *** indicating a *p*-value: 0.0001 < *p* < 0.001.

## Data Availability

Data is contained within the article and Appendix A.

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
