# Peer review of "Preparation of Bispecific IgY-scFvs Inhibition Adherences of Enterotoxigenic Escherichia coli (K88 and F18) to Porcine IPEC-J2 Cell"

_ijms, 2024, doi:10.3390/ijms25073638_

Round 1

Reviewer 1 Report

Comments and Suggestions for Authors

The paper presents a novel approach to combating Enterotoxigenic Escherichia coli (ETEC)-induced diarrhea in piglets. The study involves the generation of bispecific IgY-single-chain variable fragments (scFvs) targeting ETEC strains K88 and F18. The developed scFvs demonstrate inhibitory effects on ETEC adhesion to intestinal cells in vitro. The findings suggest a potential avenue for the development of scFv-based drugs to treat and prevent ETEC-induced diarrhea, introducing a promising strategy for addressing this prevalent issue in pig farming. The paper demonstrates a robust experimental design and meticulous methodology in generating bispecific IgY-scFvs targeting ETEC strains. The clarity of presentation enhances the overall quality of the manuscript.

The following minor suggestions are proposed for the authors' consideration to enhance clarity, provide additional context, and guide the reader through the research:

Line 12. After the 1st sentence of the abstract, consider providing a bit more context on the prevalence of ETEC K88 and F18 to enhance the reader's understanding.

Lines 35. Give data on the prevalence of ETEC K88 and F18 as the cause of Postweaning diarrhea (PWD) in piglets.

Lines 51-77. Considering the extensive length of the introduction, I would propose that this section could be condensed. This section provides valuable information, but a more concise presentation might be achieved without compromising the clarity of the research rationale.

Line 122. Provide a brief rationale for choosing the specific two and two clones for further testing.

Lines 167-184. Include a brief discussion on the observed reduction in adhesion rates and its potential significance in the context of ETEC infection control.

Lines 205-213. It would be beneficial to briefly discuss how the identified differences in CDR regions align with the varied inhibitory effects observed on ETEC adhesion.

Lines 236-257. Include a brief statement on the choice of using formaldehyde for bacterial inactivation and how this method compares to other commonly used inactivation methods.

Lines 276-295. Provide a brief explanation of why the defined IgY titer threshold (OD sample/OD negative >2.1) was chosen.

Lines 334-344. Include a short statement on why BL21DE3 cells were chosen for scFv expression and if there were any considerations in selecting this strain.

Lines 381-385. Consider mentioning any limitations or challenges encountered during the study to provide a balanced perspective.

Author Response

We would like to express our sincere thanks for your positive response and comments regarding our manuscript above. We extend our special gratitude to the reviewers for their substantial amount of time looking over the paper and their valuable comments. We found that all comments were very helpful and constructive.

Q1. Line 12. After the 1st sentence of the abstract, consider providing a bit more context on the prevalence of ETEC K88 and F18 to enhance the reader's understanding.

Answer: Thank you for your suggestion, we have added prevalence data for ETEC K88 and F18 to the abstract of the manuscript: " Enterotoxigenic Escherichia coli (ETEC) strains are significant contributors to postweaning diarrhea in piglets. Of the ETEC causing diarrhea, K88 and F18 accounted for 92.7%.".

Q2. Lines 35. Give data on the prevalence of ETEC K88 and F18 as the cause of Post-weaning diarrhea (PWD) in piglets.

Answer: Thank you for your suggestion, but we have not yet collected data on the proportion of piglet diarrhea caused by ETEC. Nevertheless, we have provided background information on the prevalence of ETEC K88 and F18 in the original manuscript: “Research conducted further underscores the significance of K88 and F18 fimbrial types, as they were detected in an overwhelming majority (92.7%) of ETEC strains associated with PWD”. We also supplemented some references on ETEC causing diarrhea in piglets [1-4].

References:

  1. Frydendahl, K., Prevalence of serogroups and virulence genes in Escherichia coli associated with postweaning diarrhea and edema disease in pigs and a comparison of diagnostic approaches. Vet Microbiol 2002, 85, (2), 169-82.
  2. Chen, X.; Gao, S.; Jiao, X.; Liu, X. F., Prevalence of serogroups and virulence factors of Escherichia coli strains isolated from pigs with postweaning diarrhoea in eastern China. Vet Microbiol 2004, 103, (1-2), 13-20.
  3. Zhang, W.; Zhao, M.; Ruesch, L.; Omot, A.; Francis, D., Prevalence of virulence genes in Escherichia coli strains recently isolated from young pigs with diarrhea in the US. Vet Microbiol 2007, 123, (1-3), 145-52.
  4. Dubreuil, J. D.; Isaacson, R. E.; Schifferli, D. M., Animal Enterotoxigenic Escherichia coli. EcoSal Plus 2016, 7, (1).

Q3. Lines 51-77. Considering the extensive length of the introduction, I would propose that this section could be condensed. This section provides valuable information, but a more concise presentation might be achieved without compromising the clarity of the research rationale.

Answer: Thank you very much for your advice, we have condensed this section in the manuscript with tracking changes.

Q4. Line 122. Provide a brief rationale for choosing the specific two and two clones for further testing.

Answer: Thank you very much for your suggestion. We have supplemented this section in the manuscript. The selection process involved choosing the top five clones with the highest ratios of K88 antigen wells to CBS wells and the top five clones with the highest ratios of F18 antigen wells to CBS wells. Among the five anti-K88 scFv clones, two distinct anti-K88 scFv sequences were identified through sequencing by the GBACK primer and categorizing based on the CDRs' amino acid sequences. Similarly, among the five anti-F18 scFv clones, two distinct anti-F18 scFv sequences were identified using the same method.

Q5. Lines 167-184. Include a brief discussion on the observed reduction in adhesion rates and its potential significance in the context of ETEC infection control.

Answer: Thank you very much for your suggestion. In the original manuscript, we discussed: Diarrhea caused by ETEC in piglets is usually not caused by one bacterial agent; therefore, therapeutic drugs that target multiple pathogens simultaneously are urgently needed. The bispecific IgY-scFv binds to both the K88 and F18 epitopes and was found to inhibit the adhesion of K88 and F18 to IPEC-J2 cells, which was comparable to the inhibitory effects observed with the anti-K88 IgY-scFv and anti-F18-scFv. To further clarify this discussion, we have supplemented some additional information: Studies have shown that IgY can protect the body from ETEC attacks by inhibiting the adhesion of ETEC to the intestinal tract[1]. This indicates that the antibody may be a potential therapeutic agent for diarrhea caused by ETEC.

References:

  1. Jin, L. Z.; Baidoo, S. K.; Marquardt, R. R.; Frohlich, A. A., In vitro inhibition of adhesion of enterotoxigenic Escherichia coli K88 to piglet intestinal mucus by egg-yolk antibodies. FEMS Immunol. Med. Microbiol. 1998, 21, (4), 313-21.

Q6. Lines 205-213. It would be beneficial to briefly discuss how the identified differences in CDR regions align with the varied inhibitory effects observed on ETEC adhesion.

Answer: Your suggestion is excellent. We had previously attempted to perform molecular docking between the antibody and ETEC antigen, but due to the complexity of the ETEC structure, this docking was unsuccessful. After reviewing relevant literature, we have added the following discussion to the manuscript: Differences in antibody sequences are evident in VH-CDR3, the region that binds tightly to the antigen[1]. Studies have demonstrated that mutations in the arginine residues of ETEC adhesins can entirely eliminate the binding activity of ETEC to host cells[2]. Therefore, modifying VH-CDR3 to enhance its interaction with the arginine residues on ETEC may be a feasible option.

References:

  1. Ohno, S.; Mori, N.; Matsunaga, T., Antigen-binding specificities of antibodies are primarily determined by seven residues of VH. Proc Natl Acad Sci U S A 1985, 82, (9), 2945-9.
  2. Baker, K. K.; Levine, M. M.; Morison, J.; Phillips, A.; Barry, E. M., CfaE tip mutations in enterotoxigenic Escherichia coli CFA/I fimbriae define critical human intestinal binding sites. Cell Microbiol 2009, 11, (5), 742-54.

Q7. Lines 236-257. Include a brief statement on the choice of using formaldehyde for bacterial inactivation and how this method compares to other commonly used inactivation methods.

Answer: Thank you very much for your suggestion. Formaldehyde, due to its low cost and effective inactivation properties, is a commonly used reagent for the preparation of inactivated vaccines. We have attached recent references[1-3] regarding the use of formaldehyde in the preparation of inactivated vaccines. Additionally, our previous studies have demonstrated that the use of formaldehyde to inactivate ETEC can elicit a strong immune response in laying hens. Therefore, in this experiment, we have chosen to continue using formaldehyde as the inactivation reagent[4].

References:

  1. Qudratullah; Muhammad, G.; Jamil, T.; Rashid, I.; Ullah, Q.; Saqib, M., Efficacy Evaluation of a Combined Hemorrhagic Septicemia-Mastitis Vaccine in Dairy Cows and Buffaloes. Animals (Basel) 2022, 12, (6).
  2. Ebrahimi-Nik, H.; Bassami, M. R.; Mohri, M.; Rad, M.; Khan, M. I., Bacterial ghost of avian pathogenic E. coli (APEC) serotype O78:K80 as a homologous vaccine against avian colibacillosis. PLoS One 2018, 13, (3), e0194888.
  3. Pavlicek, D.; Krebs, J.; Capossela, S.; Bertolo, A.; Engelhardt, B.; Pannek, J.; Stoyanov, J., Immunosenescence in persons with spinal cord injury in relation to urinary tract infections -a cross-sectional study. Ageing 2017, 14, 22.
  4. Han, S.; Yu, H.; Yang, F.; Qiao, S.; He, P., Effect of dietary supplementation with hyperimmunized hen egg yolk powder on diarrhoea incidence and intestinal health of weaned pigs. Food Agric. Immunol. 2019, 30, (1), 333-348.

Q8. Lines 276-295. Provide a brief explanation of why the defined IgY titer threshold (OD sample/OD negative >2.1) was chosen.

Answer: Thank you for your inquiry. The threshold value of 2.1 is commonly used to define antibody titers, but we have not been able to find literature explaining its origin. Nevertheless, we have attached recent references that use 2.1 as the threshold value for defining antibody titers[1-2].

References:

  1. Xiang, K.; Kusov, Y.; Ying, G.; Yan, W.; Shan, Y.; Jinyuan, W.; Na, Y.; Yan, Z.; Hongjun, L.; Maosheng, S., A Recombinant HAV Expressing a Neutralization Epitope of HEV Induces Immune Response against HAV and HEV in Mice. Viruses 2017, 9, (9).
  2. Brumfield, K.; Seo, H.; Idegwu, N.; Artman, C.; Gonyar, L.; Nataro, J.; Zhang, W.; Sack, D.; Geyer, J.; Goepp, J., Feasibility of avian antibodies as prophylaxis against enterotoxigenic escherichia coli colonization. Immunol. 2022, 13, 1011200.

Q9. Lines 334-344. Include a short statement on why BL21DE3 cells were chosen for scFv expression and if there were any considerations in selecting this strain.

Answer: Thank you very much for your suggestion. The selection of BL21DE3 cells for scFv expression is based on two primary reasons. Firstly, there exists a suppressible amber codon (TAG) between the scFv gene and the phage pIII coat protein gene in the pcomb3xss vector. In the non-suppressor strain BL21DE3, protein translation terminates at the amber codon TAG, resulting in the production of scFv antibody protein that is not fused with the pIII coat protein. Secondly, numerous studies have utilized BL21DE to generate soluble scFv. We have attached the relevant references for your perusal [1-2].

References:

  1. Zhang, F.; Chen, Y.; Ke, Y.; Zhang, L.; Zhang, B.; Yang, L.; Zhu, J., Single Chain Fragment Variable (scFv) Antibodies Targeting the Spike Protein of Porcine Epidemic Diarrhea Virus Provide Protection against Viral Infection in Piglets. Viruses 2019, 11, (1).
  2. Zhang, F.; Chen, Y.; Ke, Y.; Zhang, L.; Zhang, B.; Yang, L.; Zhu, J., Single Chain Fragment Variable (scFv) Antibodies Targeting the Spike Protein of Porcine Epidemic Diarrhea Virus Provide Protection against Viral Infection in Piglets. Viruses 2019, 11, (1).

Q10. Lines 381-385. Consider mentioning any limitations or challenges encountered during the study to provide a balanced perspective.

Answer: Thanks very much for your suggestions. we have reorganized the research work in the manuscript with tracking changes. In conclusion, bispecific IgY-scFvs were generated in Pichia pastoris strain X33 to target ETEC K88 and F18. The scFv was found to inhibit ETEC adhesion to small intestinal epithelial cells in vitro. By targeting and impeding the initial adhesion step of ETEC pathogenesis, this antibody-based intervention holds promise as a potential alternative to antibiotics, thereby mitigating the risks associated with antibiotic resistance and residual drug contamination in livestock production. Overall, these findings could lead to the development of scFv-based drugs to treat and prevent ETEC-induced diarrhea.

Reviewer 2 Report

Comments and Suggestions for Authors

1.Line 45: While the authors assert that "antibiotics are currently the mainstay for treating bacterial infections," the citation provided is from the year of 2000. Considering the significance of this study, it is advisable to reference a more recent publication (PMID: 30157463) addressing antibiotic resistance.

2.Despite the clear presentation of results, incorporating negative control data for each experiment, as demonstrated in figures 1, S1, and S3, would enhance the comprehensiveness of the findings.

3.Line 122: The authors should provide a rationale for selecting these scFvs against K88 or F18. Additionally, specify the origin of the scFv clones mentioned in Line 117 (96 clones were measured).

4.Line 138: Add "(Figure. 4C and 4D)" after "scFv can inhibit the adhesion of ETEC to IPEC-J2" for clarity.

5. The authors should elucidate why the IgY-scFv expressed in BL21(DE3) has a molecular weight of approximately 26 kDa, whereas in P. pastoris, it is 61 kDa. Both variants possess a His-tag, prompting consideration of potential glycosylation differences.

6. Line 378: Clearly specify the data analysis method employed. For instance, indicate whether statistical methods such as the Student's t-test were utilized.

7.It is recommended to conduct experiments evaluating the in vivo effectiveness of these IgY-scFvs against E. coli infection. Or, at least, explore the potential growth inhibitory effects of these IgY-scFvs on E. coli in vitro.

Comments on the Quality of English Language

Minor editing of English language required. Such as line 135-138, add cohesive words to enhance readability.

Author Response

We would like to express our sincere thanks for your positive response and comments regarding our manuscript above. We extend our special gratitude to the reviewers for their substantial amount of time looking over the paper and their valuable comments. We found that all comments were very helpful and constructive. 

Q1. Line 45: While the authors assert that "antibiotics are currently the mainstay for treating bacterial infections," the citation provided is from the year of 2000. Considering the significance of this study, it is advisable to reference a more recent publication (PMID: 30157463) addressing antibiotic resistance.

Answer: Your suggestions are greatly appreciated and we have made additional references to the manuscript.

References:

  1. Gao, J.; Duan, X.; Li, X.; Cao, H.; Wang, Y.; Zheng, S. J., Emerging of a highly pathogenic and multi-drug resistant strain of Escherichia coli causing an outbreak of colibacillosis in chickens. Infect Genet Evol 2018, 65, 392-398.

Q2. Despite the clear presentation of results, incorporating negative control data for each experiment, as demonstrated in figures 1, S1, and S3, would enhance the comprehensiveness of the findings.

Answer: Thank you very much for your suggestion. We have supplemented Figure S1 with PCR images of strains lacking fimbrial genes. Figures 1 and S3 are essential steps in the library construction process, and our objective is to obtain the correct PCR bands. Therefore, we have not been able to provide negative controls for these figures.

Q3. Line 122: The authors should provide a rationale for selecting these scFvs against K88 or F18. Additionally, specify the origin of the scFv clones mentioned in Line 117 (96 clones were measured).

Answer: Thank you very much for your reminder, we have made the necessary supplements in the manuscript. The selection process involved choosing the top five clones with the highest ratios of K88 antigen wells to CBS wells and the top five clones with the highest ratios of F18 antigen wells to CBS wells. Among the five anti-K88 scFv clones, two distinct anti-K88 scFv sequences were identified through sequencing by the GBACK primer and categorizing based on the CDRs' amino acid sequences. Similarly, among the five anti-F18 scFv clones, two distinct anti-F18 scFv sequences were identified using the same method. Additionally, the selected clones were annotated on the figures.

Q4. Line 138: Add "(Figure. 4C and 4D)" after "scFv can inhibit the adhesion of ETEC to IPEC-J2" for clarity.

Answer: Thank you very much for your reminder, we have made the necessary supplements in the manuscript.

Q5. The authors should elucidate why the IgY-scFv expressed in BL21(DE3) has a molecular weight of approximately 26 kDa, whereas in P. pastoris, it is 61 kDa. Both variants possess a His-tag, prompting consideration of potential glycosylation differences.

Answer: Thank you for your suggestion. We have supplemented the results section of the manuscript to address the increase in protein size due to glycosylation in Pichia pastoris, and have attached references that support the larger molecular weight of glycosylated single-chain antibodies.

References:

  1. Royle, K. E.; Polizzi, K., A streamlined cloning workflow minimising the time-to-strain pipeline for Pichia pastoris. Rep. 2017, 7, (1), 15817.

Q6. Line 378: Clearly specify the data analysis method employed. For instance, indicate whether statistical methods such as the Student's t-test were utilized.

Answer: Thank you for your reminder. We have supplemented the data section with the following information: "One-way analysis of variance (ANOVA) and Student's t-test were used to compare the results."

Q7. It is recommended to conduct experiments evaluating the in vivo effectiveness of these IgY-scFvs against E. coli infection. Or, at least, explore the potential growth inhibitory effects of these IgY-scFvs on E. coli in vitro.

Answer: Your suggestion is excellent. While in vivo experiments and growth inhibition tests are the best validations, they require a large quantity of bispecific single-chain antibodies, which is currently unfeasible given our limited purification capabilities. In subsequent studies, we will optimize fermentation conditions to prepare the antibodies in large scale and then conduct the aforementioned experiments.Meanwhile, the focus of this paper is on the Preparation of bispecific IgY-scFvs inhibition adherences of enterotoxigenic Escherichia coli (K88 and F18) to porcine IPEC-J2 cells. Research has shown that the inhibitory effect of IgY on ETEC is mainly attributed to its anti-adhesive properties [1,2], and IPEC-J2 cells are a well-established model for studying ETEC adhesion [3,4]. Therefore, based on these viewpoints, we have validated the adhesive inhibition capability of bispecific single-chain antibodies in vitro in this paper.

References:

  1. Jin, L. Z.; Baidoo, S. K.; Marquardt, R. R.; Frohlich, A. A., In vitro inhibition of adhesion of enterotoxigenic Escherichia coli K88 to piglet intestinal mucus by egg-yolk antibodies. FEMS Immunol. Med. Microbiol. 1998, 21, (4), 313-21.
  2. Xu, Y.; Li, X.; Jin, L.; Zhen, Y.; Lu, Y.; Li, S.; You, J.; Wang, L., Application of chicken egg yolk immunoglobulins in the control of terrestrial and aquatic animal diseases: a review. Adv. 2011, 29, (6), 860-8.
  3. Duan, Q.; Pang, S.; Wu, W.; Jiang, B.; Zhang, W.; Liu, S.; Wang, X.; Pan, Z.; Zhu, G., A multivalent vaccine candidate targeting enterotoxigenic Escherichia coli fimbriae for broadly protecting against porcine post-weaning diarrhea. Vet Res 2020, 51, (1), 93.
  4. Brosnahan, A. J.; Brown, D. R., Porcine IPEC-J2 intestinal epithelial cells in microbiological investigations. Vet Microbiol 2012, 156, (3-4), 229-37.

Round 2

Reviewer 2 Report

Comments and Suggestions for Authors

Thanks to the author for diligently revising the manuscript. The revised version evinces commendable improvement, indicating its readiness for publication.